# Paradigms and Scenarios for the Dark Matter Phenomenon

**Paolo Salucci [1,2,*], Nicola Turini [3,4]**  **and Chiara di Paolo [1,2]**

1   Sissa, Via Bonomea 262, 34136 Trieste, Italy; cdipaolo@sissa.it
2   INFN-TS Iniziativa Specifica QGSKY, Via Valerio, 2-34127 Trieste, Italy
3   Physics Department, University of Siena, Via Roma 56, 53100 Siena, Italy; nicola.turini@cern.ch
4   INFN Gruppo collegato di Siena, Via Roma 56, 53100 Siena, Italy
*   Correspondence: salucci@sissa.it

**Abstract:** Well known scaling laws among the structural properties of the dark and the luminous matter in disc systems are too complex to be arisen by two inert components that just share the same gravitational field. This brings us to critically focus on the 30-year-old paradigm, that, resting on a priori knowledge of the nature of Dark Matter (DM), has led us to a restricted number of scenarios, especially favouring the collisionless Λ Cold Dark Matter one. Motivated by such observational evidence, we propose to resolve the dark matter mystery by following a new Paradigm: the nature of DM must be guessed/derived by deeply analyzing the properties of the dark and luminous mass distribution at galactic scales. The immediate application of this paradigm leads us to propose the existence of a direct interaction between Dark and Standard Model particles, which has finely shaped the inner regions of galaxies.

**Keywords:** dark matter; galaxies; cosmology

## 1. Introduction

The mass distribution in Spirals is largely dominated by a dark component as it is evident from their kinematics and their other tracers of the mass distribution (e.g., see [1]). More in general, many other observations indicate the presence of such "substance" in the Universe. Among those, the gravitational lensing of background objects, the extraordinary Bullet Cluster [2], the temperature distribution in Clusters of galaxies (e.g., [3]) and, more recently, the pattern of anisotropies in the cosmic microwave background (CMB) radiation ([4]). Furthermore, the theory of Big Bang nucleosynthesis indicates that the vast majority of dark matter in the Universe cannot be made by baryons. With the caveat of an (exotic) population of primordial Black Holes, the Dark Matter is therefore thought to be made of massive particles that interact with Standard Model particles and with themselves mainly via Gravitation: the non-gravitational interactions are believed to have cross sections very small (for WIMPS: $10^{-26}$ cm$^2$ ) and no role in the building of the cosmological structures. Noticeably, the current belief is that such DM-Luminous Matter (hereafter LM) interactions provide us with messengers of the dark particle.

In the past 30 years, the leading approach to the 'DM mystery' has not been astrophysical or experimental but has followed a particular route that in Physics has often been successful. Everything starts by adopting the *Paradigm* according to which strong theoretical arguments on how nature could be made lead us to the correct cosmological scenario and, in turn, to the actual dark particle in which the detectability via experiments and astrophysical observations results as a bonus of the same arguments above. This Paradigm has pointed especially to a stable Weakly Interacting Massive Particle (WIMP), likely coming from SuperSymmetric extensions of the Standard

Model of Elementary Particles [5,6] and has opened the way for the collisionless ΛCDM scenario. In spite of a good agreement of its predictions with many cosmological observations, at galactic scales, the above scenario runs in serious problems including the well known one for which the predicted structural properties of DM halos result in strong disagreement with respect to those inferred from the internal motions of galaxies (see, e.g., [1]). It has been claimed that these strong discrepancies could be eliminated by astrophysical processes (e.g., [7]) in which supernovae explosions eventually flatten the originally cusped DM density profiles, however, as new data come in, the DM halos density profiles appear to be always more difficult to be accounted by such processes (e.g., [8,9]). As an example of this, the presence of very large DM halo core radii in Low Surface Brightness galaxies [8]. Furthermore, it is important to stress that, despite the large efforts made in searching for them, the WIMP particles have not turned up in direct, indirect and LHC collider searches (see, e.g., [10,11]).[1]

Consequently, being bound to the goal of resolving and framing the 'Dark Matter Phenomenon', these arguments and others that we present in this work for the currently leading scenario, motivate us to come back to the starting point and to propose a new *Paradigm* and to follow its directive towards a new *scenario*.

The plan of this work is the following: in the next section we will lead the reader, also by presenting further evidences, towards our new Paradigm for Dark matter. In Section 3 we will use the latter to work out a scenario for the Dark Matter Phenomenon, in which the most relevant predictions are checked in the following section. In the next section we will discuss the results obtained in this work also in light of the issues left in the previous sections for further deepening. In the last Section we will draw our conclusions.

## 2. The New Paradigm: Motivations and Statements

From individual and coadded rotation curves of Spirals we can obtain their mass distribution (see Appendix A and [1,12] for details). Their structural mass components include the well-known exponential stellar disc, with surface density profile [13]:

$$\mu(r; M_D) = \frac{M_D}{2\pi R_D^2} \, e^{-r/R_D} \tag{1}$$

where $R_D$ is the stellar disc scale length derived from galaxy photometry and $M_D$ is the stellar disc mass. We obtain $\rho_\star(r; M_D)$, the stellar density, by assuming that the stellar disk has a thickness of $0.1 \, R_D$, as found from the photometry of edge-on Spirals, then:

$$\rho_\star(r; R_D, M_D) = \frac{\mu(r; R_D, M_D)}{0.1 R_D} \tag{2}$$

The DM halo component is assumed to follow the cored Burkert halo profile ([1,14]):

$$\rho_B(r; r_0, \rho_0) = \frac{\rho_0 r_0^3}{(r + r_0)(r^2 + r_0^2)} \tag{3}$$

with $\rho_0$ the DM halo central density and $r_0$ the DM halo core radius. The velocity model reads as:

$$V_{mod}^2(r; r_0, \rho_0, M_D) = V_B^2(r; r_0, \rho_0) + V_D^2(r; M_D) \tag{4}$$

with: $V_D^2(y; M_D) = \frac{GM_D}{2R_D} y^2 Be\left(\frac{y}{2}\right)$ where $y \equiv r/R_D$, $G$ is the gravitational constant and $Be = I_0 K_0 - I_1 K_1$ is a combination of Bessel functions and with: $V_B^2(r; r_0, \rho_0) = 6.4 \frac{\rho_0 r_0^3}{r} (\ln(1 + \frac{r}{r_0}) - \arctan(\frac{r}{r_0}) +$

---

[1] It is worth to point out that the same problems also occur for the others scenarios that, alongside with the leading ΛCDM one, arise from the above Paradigm.

$\frac{1}{2} \ln(1 + \frac{r^2}{r_0^2})$). The model fits extremely well all the kinematics, individual and coadded, of the disk systems. For Normal Spirals, it has three free parameters that, alongside with the observational quantity $R_D$, emerge all as specific functions of the galaxy luminosity in the I band which, in turn, results as a function of the halo virial mass $M_{vir}$ (see Equations (6a)–(10) in [12] and Figure A1 in Appendix A). Defining $M_{DM}(r)$ a generic DM mass profile, $M_{vir}$, for a Burkert DM profile is given by (cgs units):

$$M_{vir} \equiv M_{DM}(R_{vir}) = \frac{4}{3} \pi \, 100 \times 10^{-29} \, R_{vir}^3 = \int_0^{R_{vir}} 4\pi\rho_B(r;\rho_0,r_0) \, r^2 dr \qquad (5)$$

We notice in Equation (3) that the DM halos have a characteristic density length scale $r_0(M_{vir})$ that separates them in two different regions: an inner one in which the dark particles are not distributed as being collisionless, and outer one in which they are likely so. With this brief review we have introduced the fact that, in Spirals, the dark and the luminous densities take the form:

$$\rho_B(r; r_0(M_{vir}), \rho_0(M_{vir})) \, , \quad \rho_\star(r; M_D(M_{vir}), R_D(M_{vir}))$$

and are known for any object of virial mass $M_{vir}$ (or of magnitude $M_I$ related to the latter with a tight relationship (see [12]))[2] .

Inspecting the derived DM–LM mass structures, the first amazing relationship that emerges features the size of the DM constant density region $r_0$ that tightly correlates with the stellar disc scale length $R_D$ (see Figure 1). We have:

$$Log \, r_0 = (1.38 \pm 0.15) \, Log \, R_D + 0.47 \pm 0.03 \qquad (6)$$

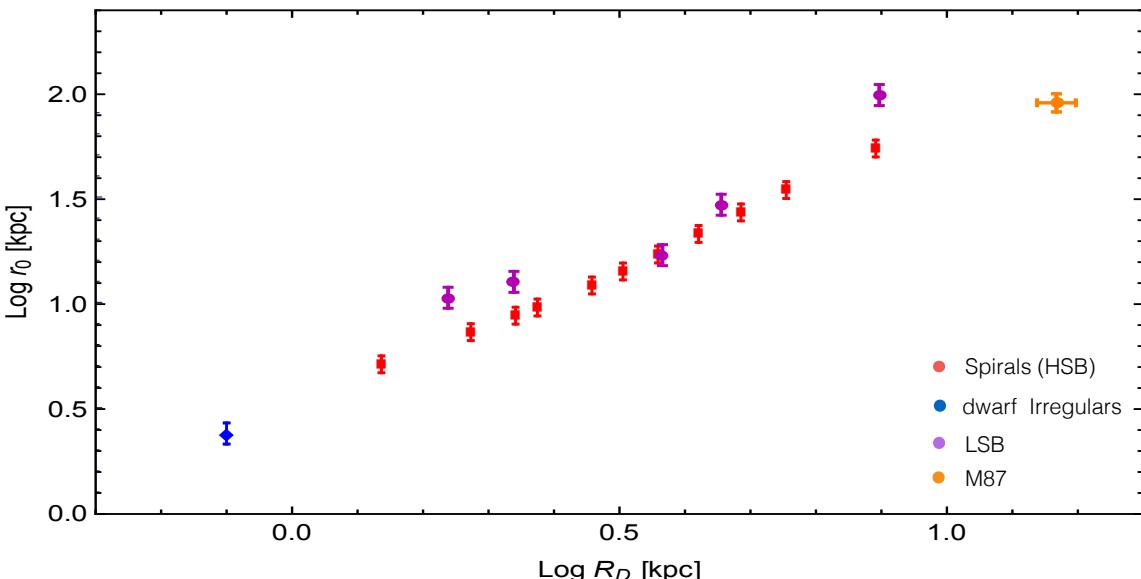

**Figure 1.** Log $r_0$ *vs* log $R_D$ in normal Spirals (*red*), dwarf Spirals (*blue*), Low Surface Brightness (*magenta*) and the giant elliptical M87 (*orange*).

This relationship, first found in [15], is confirmed today in 2300 Spirals [12,16], in 72/36 LSB and Dwarf Irregulars and in the giant cD galaxy M87 [8,9]. Overall, the relationship extends over three orders of magnitudes in galaxy luminosity. Noticeably, the quantities involved, $r_0$ and $R_D$, are derived

---

[2]    For simplicity of writing, sometime, in the quantity $r_0$ we do not explicit its $M_{vir}$ dependence.

in totally independent ways: by accurate modelling of the galaxy kinematics the one, and by fitting the galaxy photometry the other.

The second feature concerns the evidence that in Spirals, at $r = r_0$, the luminous and the dark surface densities: $\Sigma_\star(r_0, M_D) \propto \frac{M_D}{4\pi R_D^2} e^{-r_0/R_D}$ and $\Sigma_B(r_0, \rho_0) \propto \rho_0 r_0$ strongly correlate (see [17]) independently of whether the inner region of a galaxy, with $r < r_0$, is DM or LM dominated.

Finally, at any fixed halo mass (or stellar disk mass), galaxies that result more (less) compact than the average in the stellar component, result also more (less) compact than the average in the dark component [8,9]. Incidentally, this relationship is another problem for the baryonic feedback scenario: more compact is the stellar distributions and *more* DM particles are removed from the original halo cusps. It is worth specifying that these three relationships are published and very well known, but they have become absolutely crucial in coping with the DM Phenomenon only after that new observational evidence has been recently added. Therefore, they are considered in such role here for the first time.

The above relationships have no evident direct justification from the currently accepted First Principles of Physics leading, via the standard Paradigm, to the DM scenarios of WIMPS, keV-neutrinos, Ultra Light Axions. Noticeably, in all of them the Dark Matter particles interact with the rest of the Universe essentially only by Gravity and therefore, for these scenarios, one cannot contemplate the existence, in the dark matter density profile, of a constant region of size $r_0$ with the property for which the LM quantities: $\rho_\star(r_0)$ and $dlog\,\rho_\star/dlog\,r\,(r_0)$ tightly correlate with the corresponding DM quantities: $\rho_B(r_0)$ and $dlog\,\rho_B/dlog\,r\,(r_0)$.

The DM halo length-scale $r_0$ has also another important and unexpected property that we put forward in this work. Let us assume for the DM halos a spherical symmetry and an isotropic pressure support. It is useful to recall that in Euler's equation for *collisionless* isotropic systems (as a DM halo of density $\rho_B(r)$) [3] with the function of balancing the variations of the gravitational potential $\frac{d\Phi_B}{dr}$, a term $-\frac{1}{3}d(\rho_B(r)V^2(r))/dr$ appears representing something akin to the pressure force $-dP(r)/dr$. More precisely, the term $\frac{1}{3}\rho_B(r)V^2(r)$ represents one of the three equal diagonal components of the Stress Tensor of the above spherical isotropic configuration [4]. Then, we introduce the pressure $P(r)$ of the collisionless (over the free-fall time) DM halo:

$$P(r; M_{vir}) = \frac{1}{3}\,\rho_B(r; M_{vir})V(r; M_{vir})^2$$

with $V(r; M_{vir}) = V_{mod}(r; r_0(M_{vir}), \rho_0(M_{vir}), M_D(M_{vir}))$ and with $Log(M_{vir}/M_\odot)$ ranging from 10.9 to 12.7 ([12]). $P(r; M_{vir})$ (see Figure 2 *up* ) is null at the galaxy centre, then, increases outwards reaching a maximum value at $r = R_{cp}$, the "constant pressure" radius where $dP/dr = 0$ and $dP/dr < 0$ when $r > R_{cp}$. Remarkably, we have: $R_{cp} \simeq r_0$ (see Figure 2 *bottom*) so, in very good approximation, $r_0$ is the radius at which the DM pressure shows a sort of discontinuity in its profile with the value $P(r_0(M_{vir}), M_{vir})$ varying less than a factor 1.5 among Spirals of different masses.

In conclusion, all this takes us to the claim that the quantity $r_0$ marks the edge of the region where significant DM–LM interactions have taken place. Hereafter, we will often adopt such a claim.

Furthermore, always in Spirals and unexpectedly in a collisionless DM scenario, the dark and luminous *densities* emerge strongly correlated. First, in analogy with the self-annihilating DM case in which the density kernel is: $K_{SA}(r) = \rho_{DM}^2(r)$ with $\rho_{DM}$ a generic DM profile, we define $K_C(r)$ as the density kernel of the DM-baryons (collisional) interaction in Spirals:

$$K_C(r) \equiv \rho_B(r)^a \rho_\star(r)^b\,v^c \tag{7}$$

---

[3]  In which the gravitational potential is measured by means of point masses in rotational equilibrium yielding the rotation curve $V(r)$.

[4]  See for details the Chapter 4 of the Binney and Tremaine book [18].

where collisional stands for absorption and/or scattering and $v$ is the relative velocity between dark and Standard Model particles. The exact form for $K_C(r)$ is unknown, however, definiteness and simplicity suggest us to assume for the processes that we consider: $a = 1$, $b = 1$ and $c = 0$. Let us stress that the kernel $K_C$ is defined on a *macroscopic* scale, i.e., it is spatially averaged over a scale of the order of the variations of the galaxy gravitational field, 1–10 kpc. On a microscopical level, where the interactions really take place, the actual kernel could be much more complex, variable and strongly dependent on the particles relative velocity.

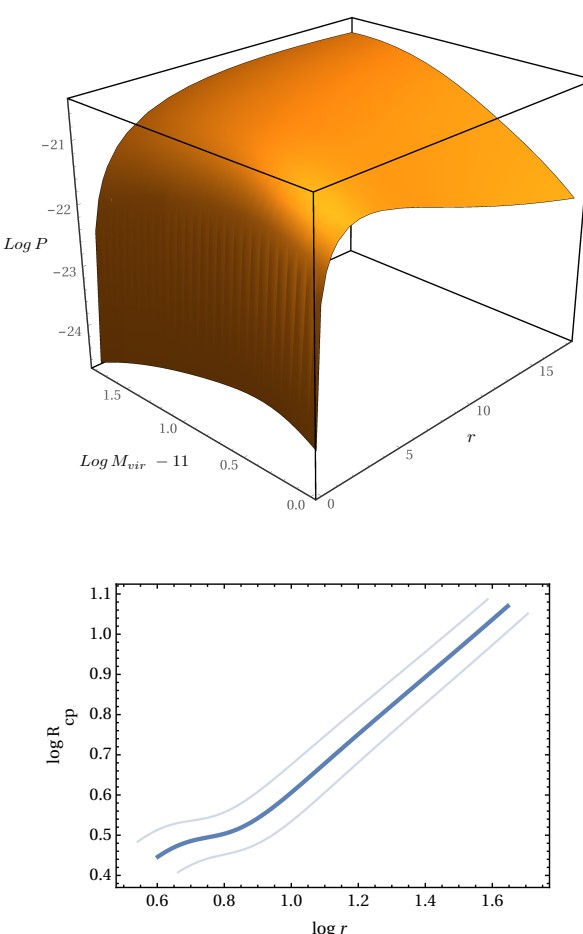

**Figure 2.** *Up.* The Dark Matter (DM) pressure in Spirals (cgs units) as function of halo mass (in solar masses) and radius (in kpc). *Bottom.* Log $R_{cp}/kpc$ *vs.* Log $r_0/kpc$.

Once we evaluate in Spirals the quantity $K_C(r_0)$, we find:

$$K_C(r_0) \simeq const = 10^{-47.5 \pm 0.3} \text{g}^2 \text{cm}^{-6} \quad . \tag{8}$$

i.e., the above kernel keeps constant within a factor of about 2, see Figure 3.

In comparison, in the same objects and at the same radii, $K_{SA}(r_0)$ varies by two orders of magnitude and an even larger variation is predicted for $K_{SA}(r_0)$ in the case of collisionless DM particles with an NFW density profile. It is also impressive (see Figure 4) that $K_C(r, M_{vir})$ varies largely both among galaxies and within each galaxy, but, only at $r \simeq r_0$, i.e., at the edge of the sphere inside which the dark-luminous matter interactions have occurred so far, takes, in any galaxy, approximately the same value of Equation (8). Such value, therefore, can be interpreted as that the minimum one for which one interaction between a dark matter particle and a Standard Model particle has taken place in about 10 Gyrs, the likely age of spiral galaxies.

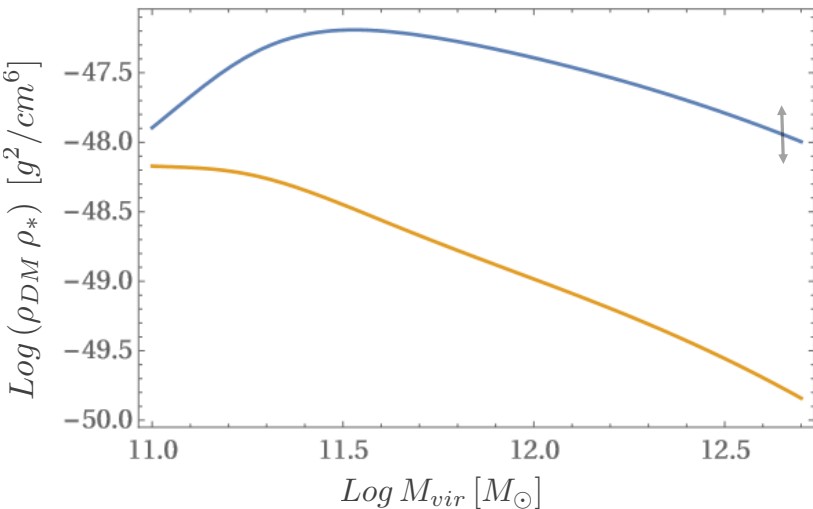

**Figure 3.** *log* $K_C(r_0)$ as a function of *log* $M_{vir}$ (*blue line*). $K_{SA}(r_0)$ is also shown (*orange line*).

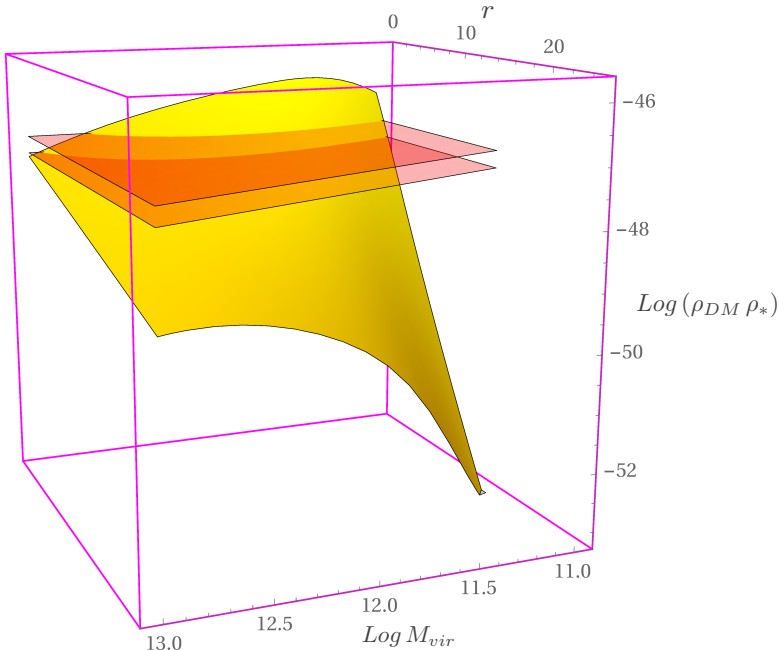

**Figure 4.** $\log K_C / (g^2 cm^{-6})$ as function of $\log M_{vir}/M_\odot$ and $r/kpc$ (*yellow surface*). In Spirals, the full range of $K_C(r_0)$ lies between the two parallel planes.

The three features reported and the three newly presented in this section, that ultimately stem from the entangled dark-luminous mass distribution in galaxies, strongly suggest that some non gravitational energy has been directly exchanged between atoms (or photons and/or neutrinos) and DM particles via processes currently unknown and seemingly not explainable within the First Principles underlying the ongoing Paradigm for the Dark Matter Phenomenon. More specifically,[5] the DM–LM entanglement in galaxies presented in previous sections works as a strong motivation for advocating a change of Paradigm, in the direction in which the nature of the dark particle and

---

5　For the first time in this work.

its related Cosmological Scenario are determined from reverse-engineering the galactic observations characterizing the DM Phenomenon.[6]

At this stage let us recall that the ongoing Paradigm starts from what we call a "First Principle" (e.g., SuperSymmetry), which highlights/introduces a specific particle (e.g., the CDM neutralino) that, in turn, yields a well defined cosmological scenario with testable predictions and applicable detection strategies (see Figure 5 for a graphical representation of this Paradigm). We stress that, even though some scholar assumes one of the above scenarios on an observational or agnostic point of view, in reality, he/she is allowed to do so only because such scenario is the outcome of a widespread 30-year-old strongly believed Paradigm.

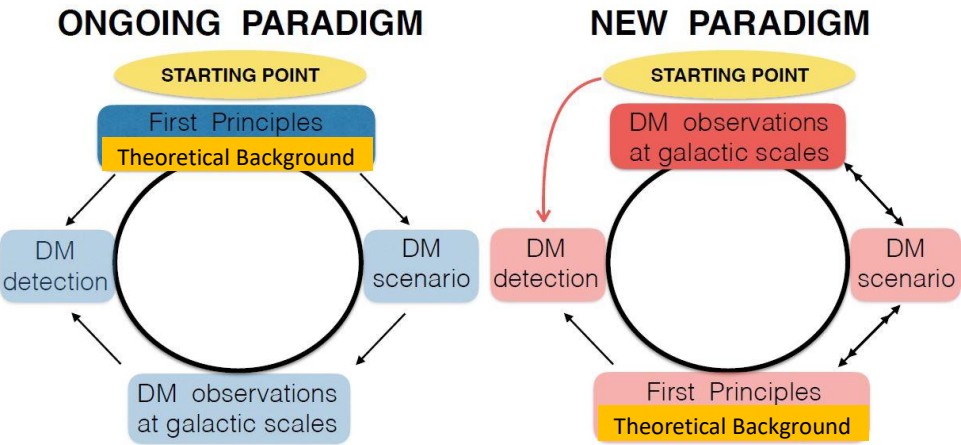

**Figure 5.** The current and the new Paradigms. Notice the different role of the galactic DM-related observations.

Our new Paradigm runs, instead, according to the following loop (see Figure 5, right): **available observations** lead us to the **DM scenario** which, once verified by other purposely planned observations, is thought to provide us with the nature of the dark particle and the **theoretical background** of the DM Phenomenon. Notice that, in principle, any scenario could come out from this Paradigm, including those with strong theoretical support, but, of course, this will depend on galactic observations. It is worth to stress that, while in this work the decisive observational evidence comes from disk galaxies for which accurate mass distributions are available, in principle, our Paradigm refers to observations relative to *any* scale and mass and to *any* virialized object of *any* morphology.

### 3. The Interacting DM Scenario

By applying the new Paradigm, a scenario immediately arises: it features dark-luminous matter interactions relevant when summed up to the age of the Universe $\simeq 10^{10}$ years and occurring in very dense regions of dark *and* luminous matter. On the other hand, on the galaxy free-fall time, at high redshifts and in the outermost halo's regions, also in this scenario, the DM particle behaves in a collisionless way. We stress that, at this starting point of the loop (see Figure 5 right), other scenarios could be compatible with the reverse-engineering of the (present) DM/LM structural properties reported above, however, in this work we will investigate the simplest one.

The particle itself is (presently) unspecified: in this respect, our scenario is open to a huge field of possibilities that should be followed up by suitable observations and experiments. However, the proposed DM particle–nucleon interactions in galaxies must have left behind a number of imprints, including the *fabrication* of the detected DM density cores and the *creation* of the emerged

---

6　　i.e., the six features discussed in this paper plus any other relevant one that should emerge.

dark–luminous matter entanglement. Let us stress that the scenario has automatically a length-scale: $R_{opt} = 3R_D$, i.e., the size of the (luminous part) of a galaxy, no DM/LM interactions can occur for $r > R_{opt}$ for an evident lack of a sufficient number of baryons (see Equation (2)). Differently from the cases of WIMPS, keV neutrino or ULA, the nature of our particle emerges only in the innermost regions of the galaxy halos. In detail, in this Interacting Dark Matter scenario, the dark halos were formed with an NFW density profile [19], which is the characteristic one for collisionless particles as ours, when time-averaged over the galaxy inner halo collapse time of $\sim 10^8 \, y$. In this formation phase, no interactions took place in the DM halos also because the baryons within these had then very low densities and no star was formed yet. Remarkably, such profile is recovered in the outermost regions of the *present day* galactic halos of Spirals; in fact, for $r > r_0$ ([12]), from the modelling of rotation curves we find: $\rho_{DM} = \rho_{NFW}$ ([12]):

$$\rho_{NFW}(r; M_{vir}) = \frac{M_{vir}}{4\pi R_{vir}} \frac{c^2 g(c)}{\tilde{x}(1 + c\tilde{x})^2} \quad , \tag{9}$$

where $R_{vir}$ is the virial radius, $\tilde{x} = r/R_{vir}$, $M_{vir}$ is the virial mass, $c \simeq 14 \, (M_{vir}/(10^{11}M_\odot))^{-0.13}$ is the concentration parameter and $g(c) = [ln(1+c) - c/(1+c)]^{-1}$ (see [12]).

This agreement, see Figure 6 in the external halo regions, between the actual and the predicted NFW profile is extraordinary since in the inner regions they totally disagree and provide us with the primordial DM halo density, once we extrapolate the RHS of Equation (9) down to $r = 0$.

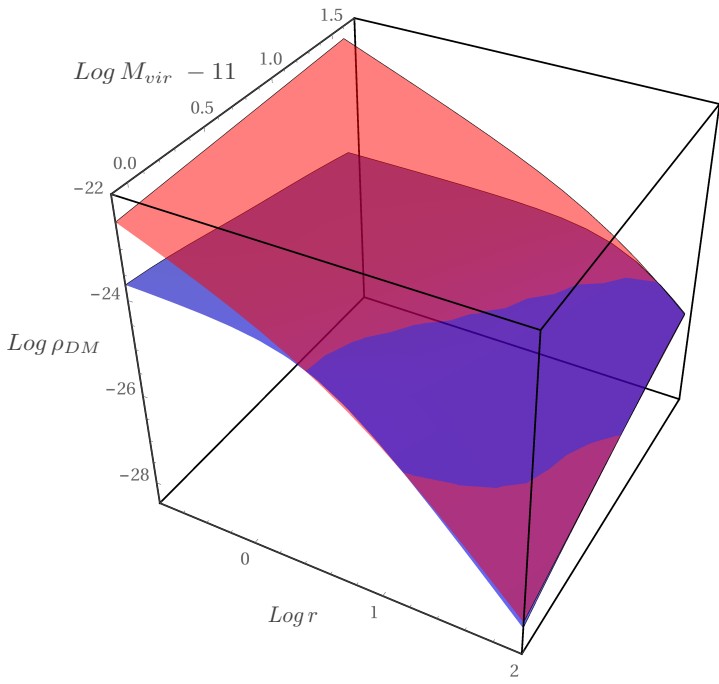

**Figure 6.** Primordial (*red*) and present-day (*blue*) dark matter density profiles in Spirals. $log \, \rho_{DM}$ (in g cm$^{-3}$) is shown as a function of log radius (in kpc) and log halo mass (in $10^{11}$ solar masses).

Then, for an object of mass $M_{vir}$, we can compute the quantity $\Delta M_{DM}(M_{vir})$, i.e., the amount of dark matter that has been *removed* from the spherical region of the DM halo of size $r_0(M_{vir})$, by the core forming interactions

$$\Delta M_{DM}(M_{vir}) = 4\pi \int_0^{r_0(M_{vir})} (\rho_{NFW}(r; M_{vir}) - \rho_B(r; M_{vir}))r^2 dr \tag{10}$$

This amount goes from 40 % to 90 % of the primordial mass of this region. Noticeably, on the other hand, in all galaxies this quantity is only 1% of the (present and primordial) total halo mass.

In a global sense, the difference between the primordial and the present DM distribution is very mild, despite the unexpected occurrence of an exchange of energy between the DM–LM particles capable of modifying the density distribution of the former over the stellar disk of galaxies. There is an important difference between our candidate particle and the previous ones: in the present case only 1% of the dark particles in galaxies ever reveal their true nature, 99% of them may be indistinguishable from a WIMP one.

Given $m_p$ the dark particle mass, the number of interactions per galaxy involved in the core-forming process is: $N_I(M_{vir}) = \Delta M_{DM}(M_{vir})/m_p$. The number of interactions for galaxy atom of mass $m_H$ is $N_{I/A}(M_{vir}) = \frac{\Delta M_{DM}(M_{vir})}{M_\star(M_{vir})} m_H/m_p$. $W(M_{vir})$, the work done during the core-forming process by flattening the primordial cusped DM density, is obtained by:

$$W(M_{vir}) = 4\pi \left( \int_0^{r_0(M_{vir})} \rho_{NFW}(r; M_{vir}) M_{NFW}(r; M_{vir}) \, r \, dr - \int_0^{r_0(M_{vir})} \rho_B(r; M_{vir}) M_B(r; M_{vir}) \, r \, dr \right) \quad (11)$$

We divide this energy by the number of interactions $N_I(M_{vir})$ taken place in each galaxy (inside $r_0(M_{vir})$ and during the Hubble time) and we get the energy per interaction per GeV mass of the dark particle (see Figure 7) :

$$E_{core} = (100 - 500) \, eV \, \frac{m_P}{GeV}$$

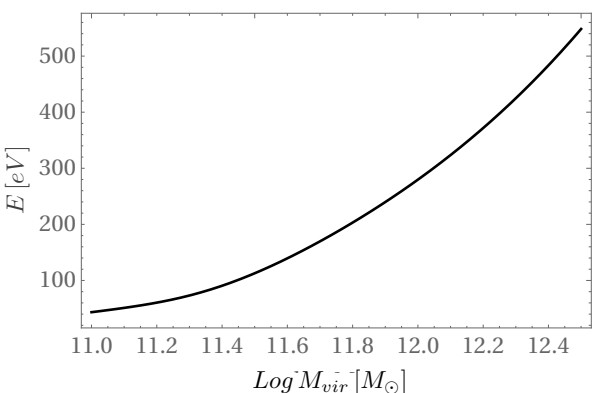

**Figure 7.** The energy of a core-forming SM–DM particles interaction for a GeV of the particle mass as a function of log $M_{vir}$.

It is important to recall that the values of the quantities above derived must be considered as averaged over the galaxy gravitational field variations scale length, i.e., several kpcs.

At a *microscopic* level, where do such interactions take place? We propose gravitationally bound objects like main sequence and/or giant stars or places with high baryonic density/temperature/relative-velocity with DM particles as white dwarfs, neutron and binary neutron stars, accretion discs to galactic black holes. Finally, the role as a source of particles interacting with the dark ones of the primordial galactic HI disk inside $r_0$, up to $10^4$ times denser than the present one has to be investigated. Noticeably, almost all the above galactic locations have a macroscopic large-scale radial distribution proportional to that of the stellar disc. It is worth to recall that stars, neutron stars and BHs are considered attractors of DM particles even in the WIMP scenario (e.g., [20,21]). In our proposed scenario we can correlate the DM particles with the SM (Standard Model) particles in three ways:

1.  Direct momentum transfer from SM to DM.
2.  DM destruction by direct interaction with SM particles (electrons or baryons).
3.  DM–DM annihilation enhanced by dense baryonic objects.

More specifically, the DM halo particles, while travelling inside the cuspy region of the DM halo, via the proposed interaction: (a) acquire, from collisions with the atoms, an extra kinetic energy $E_{core}$ sufficient to leave the region or (b) lose an amount $\sim E_{core}$ of their kinetic energy by collisions or absorption and then are captured, for example, by the stars in the region becoming *disc* particles. In both cases, it is immediate to recognize that these interactions can occur only out to a radius $r \sim 3R_D$, inside which there is a sufficient number of SM particles, setting in this way the observed entanglement between the DM/LM distributions.

Alternatively, the presence, in the primordial cuspy region, of dense baryonic objects like stars and BHs, could strongly enhance the DM self annihilation due the significant increase of the DM (galactic) density around to these $10^{9-11}$ objects and *in Hubble time scale* deplete of DM particles the inner regions of galaxies. This process creates, as in the previous cases, density cores strongly tied to the LM distribution, differently from the similar standard Self Interacting Dark Matter scenario which is blind to the distribution of LM in Galaxies.

At a *microscopic* level, the densities and temperature/relative velocities that are involved in the interaction depend on the distance from the center of the *object* where the interaction takes place. Notice that, during the whole history of the Universe, in each spiral at $r > r_0$, the (number) density of stars and the DM halo density have always been so small that the number of the corresponding dark-luminous particle interactions $N_{dlout}$ has been negligible, i.e., today: $\int_{r_0}^{r} 4\pi \, N_{dlout} \, m_P \, R^2 t_H \, dR <<$ $(M_B(r) - M_B(r_0)) \simeq (M_{NFW}(r) - M_{NFW}(r_0))$, with $t_H$ the Hubble time $10^{10}$ years. Of course, we can detect the presence of the (rare) interactions occurring also outside $r_0$ but they are core-forming and entanglement achieving only inside $r_0$.

Let us stress that in our scenario the DM–DM annihilation or the DM–SM interaction are meant to be at a very short range. In this way we preserve the $\Lambda$CDM good agreement at large scales and explain the discrepancies that we observe in the central regions of galaxies. In fact, when dark halo and stellar disk densities are extremely low and their (macroscopically spatially averaged) product is lesser than $10^{-47}$ g$^2$/cm$^6$, like in the outermost regions of galaxies in the intergalactic space and in Clusters [7] the DM is virtually collisionless. Differently, in the central regions of galaxies with $r < r_0$, the number of "dense objects" and the DM density become large enough to trigger the interaction that almost totally "eliminate" in $10^{10}$ years the residing dark particles.

By summarizing, in our scenario the relevant core-forming process works as the following: over a Hubble time a large fraction of *halo* dark particles inside (the present-day core radius) $r_0(M_{vir})$, originally distributed according to a $r^{-1}$ cusp, get displaced from their original locations until the latter is erased. Contrary to any relevant DM process introduced so far, our process occurs in secluded places where the ambient energy and angular momentum well overcome those of the interacting dark particle.

## 4. Discussion

Given the amazing results and the claims of this paper presented in the previous sections, it is worth deepening the following points:

- It is worth noticing that also in the case in which the Cosmological $\Lambda CDM$ Concordance Model itself is thought to be inadequate (e.g., [22]) we can apply our Paradigm and, by reverse-engineering the relevant Small/Large Scale observations, define a scenario for DM.
- The observational results put forward in this paper arise from the knowledge of the quantities: $\rho_B(r; M_{vir})$, $V_{mod}(r; M_{vir})$ and $\rho_\star(r; M_{vir})$ or of the same quantities but a function of radius $r$ and of: a) $M_I$ or b) $V_{opt}$. It is worth to remind that these quantities, crucial to investigate the galaxy dark-luminous matter entanglement, have been the (published and well known) outcome of a

---

[7]   The intracluster medium with a density of $\sim 10^{-28}$ g/cm$^3$ seems unable to trigger significant interactions. A deeper investigation is however needed.

longstanding project aimed to determine the mass distribution in disk systems from both their *individual* and *coadded* kinematics. In the text and in Appendix A we give a short summary of this (the interested reader can see the original papers [1,8,9,12,14,16,23,24]).

- In this work the crucial observational evidence, except that for the $log\ R_D$ vs $log\ r_0$ relationship that appears to be universal in galaxies, comes from Normal Spirals. Presently, only for this family we have a statistically sufficient number of accurate LM–DM mass profiles to allow the emergence of their entanglement. The next step, obviously, will be to gather such data for Ellipticals and Dwarf galaxies.

- The core formation is not the outcome of only our Interacting Dark Matter scenario, it could occur in scenarios with Axions DM, where the galactic structure is tied to the quantum mechanics, characteristics of this hyper-light particle, or with KeV mass collisionless fermions, like the elusive Sterile Neutrinos, responsible of a warm gas that due to their fermionic nature and the specific value of the mass cannot form the centrally cusped DM distribution, typical of the collisionless particles (see [12]). However, as in the ΛCDM + baryonic feedback one, these scenarios do not have an explanation for the DM–LM relationships introduced in this paper, that, on the other hand, result in a straightforward outcome of the process in which the dark and the SM particles interact by removing the dark particles from the inner regions of halos.

- Let us reiterate that Field Effective Theories and SuSy extensions have been developed to account for a DM sector that has a portal to SM particles through various fields. These theories are purposely built to obey to the Gauge invariance and to agree with LEP/Tevatron/LHC precise measurements on EW–QCD interactions. Then, let us notice a substantial difference between the particle in our scenario and the cold DM candidates that arise from Standard Model extensions, i.e., massive particles that are "Feebly" or "Weakly" (WIMP), interacting with SM particles. In our case, in fact, protected by the new Paradigm, the particle mass scale and its coupling constants could be blended accurately such that one could cook DM scenarios without violating precise measurements, and create an environment, compatible with the astrophysical scenario we describe, in which the production cross sections are yet not negligible but the detection at LHC is almost impossible due to the large QCD background.

- LHC could be almost blind with its standard searches if the DM sector links to SM particles through different channels that are not directly linked to quarks and QCD bosons. In fact, a large Missing Transverse Energy is the main tag in the production of invisible particles, this requires that a large transverse momentum is given to the invisible particles through efficient processes such as Drell-Yan (quark–quark to Vector/scalar mediator) or from gluon–gluon fusion [25]. If the mass hierarchy is such that the transverse momentum of the created invisible particle is soft, due to the kinematics of the decays, QCD background will mask any detection efficiently. Our new Paradigm shields this occurrence from being considered fine-tuned.

- What is the relation between current underground DM searches and our scenario? Field Effective Theories have focused on mediators that couple to quark/gluon because these are the main production sources for LHC searches [26,27]. One of the main welcomed consequences, foreseen by all these theories and SuSy, is the possibility to have a scattering between the DM candidates and the nucleus of the atoms [28]. The scattering is seen as a collective recoil of the nucleus respect to the massive DM particle with an extremely low cross section. The main stream for DM direct search in underground experiments is based on it, and it is well known that huge underground detectors search for such a specific signature. Moreover, an annual modulation in the event rate as a consequence of Earth rotation around the Sun could be put in correlation with any detected excess. This feature enhances, in certain conditions, the global sensitivity of the experiments. If we change the scenario and make the quarks and QCD bosons not directly coupled to the DM sector,

the mediators would couple the DM only to the EW bosons, or simply to photons [8]. Then, in this case it is unlikely that we can induce a nucleon scattering detectable by the current underground experiments. All the direct searches would be, therefore, blind.

- We are not the only ones to consider a change of Paradigm. The theoretical picture on the Dark Matter Phenomenon is very fluid at the moment, people are looking to extremely light Axions up to TeV Axion Like Particles (ALP), others foresee heavy vector bosons that are mediators between SM particles and the Dark sector. We have, again, feeble interacting DM particles or extremely strong self interacting DM particles. A particular case is the Scenario of Strong Self Interacting Particles. It is interesting in our view because extremely dense objects, like stars, neutron stars or BH accretion disks, could in principle become a location where the DM annihilation is enhanced by orders of magnitude because of its high density. This could mimic the not completely collisionless scenario we are envisaging here despite the lack of a direct interaction. To find the actual channel via which dark and Standard Model particle talk needs accurate investigations on dense/compact objects and on their distributions in galaxies.

- In the scenarios from the ongoing Paradigm, the observed relic DM density plays a fundamental role in shaping the theoretical background of the dark matter particle. Our Paradigm leads to scenarios that instead are agnostic to this quantity, an exception being the third case in the previous section, where we can cook easily a WIMP-like relic density run mechanism, with a new short range force between the DM particles filling up the DM sector in the early Universe. In short: under the wings of our new Paradigm the DM Phenomenon could be surprisingly different from the prevalent present ideas and explainable in a way such that "the relic density requirement" gets strongly de-potentiated.

- In recent years, a number of papers, although without having a) the support of the strong and amazing observational evidence put forward in this work and b) its paradigm-changing ambitions, can, however, be considered its precursors. To report this is beyond the present goal and will be the aim of a forthcoming paper [29].

- Primordial BHs are becoming always more popular as game-changer SM candidates [30]. Not anymore a "new" particle with "new" physics but good old SM particles with instead a very peculiar formation scheme. It is then fair to briefly explain why we think that they, instead, are likely out of the game. From a cosmological point of view, the SM Higgs field with its crucial metastability that is routinely invoked for their formation during the inflation period, is failing to explain the main features of the observed Universe. Moreover, such Higgs metastability can be "cured" by new scalar fields that curiously can instead become a main portal [31] to a DM particles sector that might include our Interactive Dark Matter (IDM) particle. On an astrophysical point of view, primordial BHs behave as a collisionless particle and are a perfect $\Lambda$CDM particle with all its issues at small scales that, in this case, cannot even attempt to be "cured" by baryonic background. Finally, the tight correlation between DM and SM in galaxies is inconceivable in this scenario.

## 5. Conclusions

The Dark Matter Phenomenon has become one of the most serious breaking points of known Physics. Specifically, in galaxies, it features strong correlations between quantities deeply-rooted in the luminous world and quantities of the dark world. These relationships, unlikely, arise from some known "First Principle" or as a result of some known astrophysical process.

This situation has lead us to propose a new *Paradigm* for the DM Phenomenon, according to which the scenario for this elusive component should be obtained from reverse-engineering the (DM-related)

---

[8]    let us recall that when a resonance at 750 GeV, decaying into photon–photon, was suspected in 2015, many different ideas on different couplings were proposed, among them a direct photon–photon coupling to new physics

observations at galactic scales [9], i.e., we claim for a new frame of mind on the role of the inferred DM properties at such scales: they should work as *motivators* not as *final verifiers* of the DM scenario.

By following this strategy, we found that in normal Spirals, the quantity $\rho_B(r)\rho_D(r)$, which is the kernel associated to the Standard Model–Dark Matter particles interaction, assumes, at the radius $r_0$, which is the edge of the DM constant density region, almost the same value in all galaxies. This opens the way for a new scenario featuring interacting DM–SM particles with a core-forming exchange of energy, i.e., the structure of the inner parts of the galaxy dark halos has been fabricated by a new Dark–Luminous interaction, which defines the nature of the dark particle.

Our scenario explains very simply the "unexplainable relationship" that has motivated the change of Paradigm (a) the size $r_0$ of the constant density region is related to the size of the stellar disk being the former directly fabricated by the latter; (b) at $r_0$ in Spirals the DM pressure is radially constant: in fact, at such radius there is an equilibrium over $10^{10}$ years between the DM halo particles eliminated inside $r_0$ and those coming from outside to compensate such a loss; (c) at $r_0$ the kernel $\rho_B(r_0)\rho_\star(r_0)$ is roughly constant: in fact, this is thought to be the value that the DM and LM particles must overcome to interact among themselves once in $10^{10}$ years; (d) being the original DM distribution very cuspy, in objects with the same halo and disk mass the more/less compact the stars are distributed, the less/more efficient the core formation process will be and the more/less compact the cored halo will result.

The IDM scenario, just emerged from the new Paradigm, is already endowed with confirmation/ detection strategies, among those:

- We expect that this particle will show up from anomalies in the internal properties of the above locations (e.g., stars).
- On galactic scales, the interaction could radiate diffuse energetic photons detectable by VHE gamma rays experiments. The Moon and the Earth atmosphere may also be a source of DM generated radiation.
- There is already today the possibility of experimental direct detection. The direct search could be completed by calorimetric measurements, for example, looking to anomalous showers development in neutrino experiments. We can in this way get evidence of DM annihilation or direct DM–SM interaction.
- If the portal to DM is mediated from scalars linked to the Higgs (Higgs portal), the detection can be obtained by studying the gamma telescopes data, like Fermi or CTA, by searching for Higgs or other likely signatures around dense objects, not excluding the Earth/Sun atmosphere.
- Recently in CMS the PPS spectrometer, detecting scattered protons from the interaction point, gave a new way to start searching for gamma–gamma induced processes. The gamma–gamma production channel has, by its nature, a very low (few GeV) transverse momentum transfer. Hence, the Met for such processes is likely to be very soft. The spectrometer for large masses (>300 GeV), allows to detect any photon induced resonance as a bump in the mass spectrum suppressing efficiently the background from proton pile-up. Although this production mechanism has a much lower ($10^{-3} - 10^{-4}$) production probability with respect to the primary QCD scattering, the large acceptance and background suppression at low momentum allows one to reach very low cross section production rates.

In any case, the present investigation of the proposed Interacting Dark Matter *scenario* has strongly confirmed the need of a Paradigm different from the ongoing one, which leads to a small set of scenarios strongly *theoretically* motivated. In our view, therefore, we should stick to the observed astrophysical facts as the tight LM–DM correlations at a galactic scale, as "ugly" and complex as they may seem from a theoretical point of view. In fact, by following the philosophical idea [32] that Beauty and the "Apollonian" underlying the presently most favoured Dark matter scenarios, may not be the

---

[9]　　and, in principle, at any other scales.

correct path to the Truth, we claim that also the "Ugliness" and the "Dionysian" of the mysterious observed dark-luminous entanglement has its potential to reach the latter.

Therefore, to expand our observational knowledge beyond that presented in this work, and reach all kind of environments starting from dwarf spheroidals and ellipticals and including Groups and Clusters of Galaxies and even certain peculiarities in the Large Scale of the Universe, should become a new primary directive before attempting any new theoretical development, including that related to the DM–SM(LM) particle couplings.

**Author Contributions:** Conceptualization, P.S. and N.T.; methodology, P.S., N.T., C.d.P.; validation, P.S., N.T. and C.d.P.; investigation, N.T., P.S., C.d.P.; data curation, C.d.P.; writing—original draft preparation, P.S., N.T.; writing—review and editing, C.d.P.; visualization, C.d.P. All authors have read and agreed to the published version of the manuscript.

**Acknowledgments:** Funding for this research comes from INFN IS qgsky.

**Conflicts of Interest:** The authors declare no conflict of interest.

## Appendix A

Here we briefly summarize the results of the works indicated in the first bullet of the Discussion section and used in order to get the goal of this work. In Figure A1 *up* we show the Universal Rotation Curve derived from a large Sample of 1000 Spirals for three different magnitudes: the normalized coadded RCs $V(R/R_{opt})/V(1)$ follow a Universal profile as a function of their normalized radius $R/R_{opt}$ and their magnitude $M_I$. The velocity data with their r.m.s are shown in Figure A1 as points with errorbars alongside with the URC model-fit (solid line). This latter involves a cored dark halo and a Freeman disk component (dotted and dashed lines). Moreover, *bottom* of the URC (i.e., the global kinematics of Spirals) implies that all the dark and luminous structural quantities are correlated: disk systems belong to just one family run by, for example, the DM halo mass (or, e.g., the galaxy Luminosity). This property holds for any disk galaxy and, very importantly, emerges also in the modelling of individual RCs.

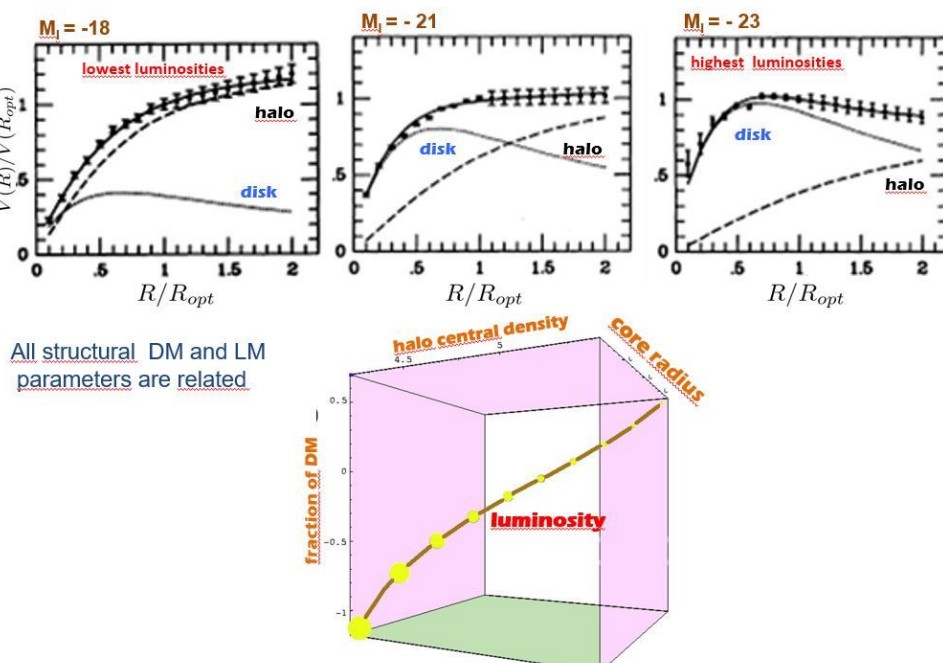

**Figure A1.** Global kinematics of Spirals and their modelling.

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
