# Peer review of "Paradigms and Scenarios for the Dark Matter Phenomenon"

_universe, doi:10.3390/universe6080118_

Round 1

Author Response

There have been misunderstatings:

For  the  statements about pressure, also collisionless systems in equilibrium do  have it. In the revised version we  better expressed this  and how this quantity arises from Euler equation and expecially we give  the  Bible-textbook reference. 

We propose a new Paradigm to investigate DM,  this could account a role for DE  but to work out a follow-up scenario  it is beyond the aim of our work.

Our results are mainly against the LCDM predictions at small scale and therefore rule out WIMPS, however, the LCDM modelization at large cosmological scale, as today, works and  it is the most credited model explaining the universe evolution. It is absolutely incorrect  to state that LamddaCDM is obsolete, in fact himself states: although, unfortunately, still prevailing in the literature.

In the present paper: we would like to refer to the previous works that are on our wavelenght , however, in the papers proposed by the referee there is not the needed spark to be  presently considered: i.e. observations at galactic scale that trigger a new DM  scenario. Any way, we are preparing a review in which we will be more open about this issue.   

Reviewer 2 Report

The authors of the manuscript Universe-872694: "Paradigms and scenarios for the dark matter phenomenon" propose a novel paradigm where the nature of dark matter (DM) should be derived by analyzing in depth the properties of dark and luminous matter distributions on galactic scales, with particular focus on certain galactic scaling relations which are to the best of current knowledge empirical. The authors then go on to suggest that the existence of these scaling relations implies the existence of direct interactions between DM and the Standard Model.

This is a very well-written paper, and in short I recommend it for publication after a very minor comment is addressed. I have noticed that the authors have not referred to any previous work which actually already follows their paradigm, proposing specific particle DM explanations for the scaling relations in question. I would recommend therefore that the authors expand Section 3 mentioning certain existing explanations in line with their paradigm.

The only example which comes to my mind to be honest is the interacting mirror or mirror-like DM model studied by Foot and collaborators, which has been over and over shown to be successful in explaining most of the scaling relations considered by the authors. Examples of works in this direction are Foot, Phys.Dark Univ. 5-6 (2014) 236-239 [arXiv:1303.1727]; Foot, Phys. Rev. D 88 (2013) 023520 [arXiv:1304.4717]; Foot & Vagnozzi, Phys. Rev. D 91 (2015) 023512 [arXiv:1409.7174]. However, I am sure other examples exist in the literature. The authors should briefly discuss and reference these possible solutions.

Author Response

We thank the referee for the comments. We have seriously considered the possibility to give in the paper  a review of previous works in  transit between DM paradigms and scenarios. However, this, to be done correctly,  would require very much time and we have decided to write a review paper  dedicated to this aim (open also to other interested!)

Reviewer 3 Report

The authors discuss a possible new direction of paradigms to account for the dark matter phenomenon. The major problem of the current dark matter paradigm discussed is the core-cusp problem. The entire discussion is not new and many related review articles are discussing this issue. 

The authors are trying to use the possible interactions between dark matter and baryons to tackle the core-cusp problem. I would say this is a possible way to solve it. However, no solid evidence is supporting this idea. The authors provide a few pieces of evidence such as the correlation between core density and disc scale length and the constant density kernel for spiral galaxies. The former one has been discussed in previous literature while the latter one is new. I would suggest the authors to discuss more about this constant density kernel. For example, is it the same for galaxy clusters or elliptical galaxies? I am dubious on whether it is also true for both dwarf galaxies and normal galaxies.

Secondly, the authors focus on the galaxies only. Any new paradigms to be considered should also account for the dark matter in galaxy clusters and cosmology. Why only focus on galaxies?

Finally, the authors suggest that the enhanced dark matter annihilation could be a possible interaction. The annihilation rate is significantly enhanced only around supermassive black holes. Also, the size of the density spike is very small (~ pc) so that the core-cusp problem cannot be easily solved by this way. 

Overall, this may be an interesting review article on the need of a new paradigm for dark matter. 

Author Response

First, in the revised version we will better explain that a main result of this paper is to claim a change of paradigm for the DM phenomenon and that enough evidences for this are already in place. We transferred the critics  to the WIMP scenario to the Paradigm that has created it.

The referee is right that all observable Universe , starting from galaxies of other Hubble Types should be involved in this  new paradigm and in its IDM related scenario. In the revised version we make clear in the conclusions that this is a future priority. 

On  suggestion of the referee we have also briefly discussed our Scenario in relation to  Clusters of galaxies. 

Finally , we better explain, in the revised version  that we do not consider the central massive BH as enhancer of the DM densities but the action of   \sim  10^10 galactic BHs  for \sim  10^10 years  in locally  enhancing the galactic DM density till to 10 order of magnitudes the galactic \sim 10^-25 g/cm^3 value . We also pointed out that  this is an idea to develop   

Round 2

Author Response

We thank the referee for sharing his views . We have no objections a priori to them. In this revised version we have 1) cited one of the paper in question 2) expressed more clearly the weaknesses of the LCDM scenario.

Round 3

Reviewer 1 Report

The authors kindly decided to cite my name in the revised version of their manuscript and I thank them for this. However, as clearly stated in my reports, I, as a referee, by no means, intended to get a citation (surely, not complete) of my work in their manuscript. Instead, from the very beginning, I, simply, thought that it would for the benefit of their article, if the authors were aware of the recent, complete and successful confrontation of the cosmological problem, nowadays. For this reason only, I gave them the detailed information and my bibliography on the "Polytropic Cosmological Model". In view of the above, I think that the authors could, if they wished so, withdraw my citation from their manuscript. If, however, they insist on citing my work, then I kindly notice that, contrary to what appears in the text (section 4 page 10, lines 240-242), the "ΛCDM Cosmological Model" should not be named "incorrect", because, in fact, the mathematical-analytical calculations in it are believed to be correct. As explained in my reports, the main deficiency (one of the many) of the "ΛCDM Cosmological Model" (as compared to the physically and mathematically oriented "Polytropic Cosmological Model") is that it is physically insufficient and inadequate, and the same holds for its misleading theoretical predictions. In this sense, the "ΛCDM Cosmological Model" cannot be termed incorrect, but, also, at the same time, it is not correct. The problem lies with the terminology used. After all, as demonstrated in one of the five publications, the "Polytropic Cosmological Model" possesses a well-defined ΛCDM-limit (namely, when the pressure and thermodynamics of the total matter are ignored), in which case all the results of the ΛCDM Cosmology are recovered. Perhaps, the authors would like to consider the above and change accordingly the text and the wording above in it.
These comments can be seen as the appendix of my last report. Many thanks.

Author Response

Dear Referee , now we have changed incorrect with

thought to be inadequate. Please do not delay the publication of the paper. Thanks